# Multidrug Resistance (MDR): A Widespread Phenomenon in Pharmacological Therapies

**DOI:** 10.3390/molecules27030616

**Published:** 2022-01-18

**Authors:** Alessia Catalano, Domenico Iacopetta, Jessica Ceramella, Domenica Scumaci, Federica Giuzio, Carmela Saturnino, Stefano Aquaro, Camillo Rosano, Maria Stefania Sinicropi

**Affiliations:** 1Department of Pharmacy-Drug Sciences, University of Bari “Aldo Moro”, 70126 Bari, Italy; alessia.catalano@uniba.it; 2Department of Pharmacy, Health and Nutritional Sciences, University of Calabria, 87036 Arcavacata di Rende, Italy; domenico.iacopetta@unical.it (D.I.); stefano.aquaro@unical.it (S.A.); s.sinicropi@unical.it (M.S.S.); 3Research Center on Advanced Biochemistry and Molecular Biology, Department of Experimental and Clinical Medicine, Magna Græcia University of Catanzaro, 88100 Catanzaro, Italy; scumaci@unicz.it; 4International PhD Programme ‘Sciences’, Department of Science, University of Basilicata, Viale dell’Ateneo Lucano n. 10, 85100 Potenza, Italy; federica.giuzio@unibas.it; 5Department of Science, University of Basilicata, 85100 Potenza, Italy; carmela.saturnino@unibas.it; 6Proteomics and Mass Spectrometry Unit, IRCCS Policlinico San Martino, Largo Rosanna Benzi, 10, 16132 Genoa, Italy; camillo.rosano@hsanmartino.it

**Keywords:** multidrug resistance, antimicrobial resistance, bacterial resistance, MDR, AMR, bacteria, antibiotics, anticancer agents

## Abstract

Multidrug resistance is a leading concern in public health. It describes a complex phenotype whose predominant feature is resistance to a wide range of structurally unrelated cytotoxic compounds, many of which are anticancer agents. Multidrug resistance may be also related to antimicrobial drugs, and is known to be one of the most serious global public health threats of this century. Indeed, this phenomenon has increased both mortality and morbidity as a consequence of treatment failures and its incidence in healthcare costs. The large amounts of antibiotics used in human therapies, as well as for farm animals and even for fishes in aquaculture, resulted in the selection of pathogenic bacteria resistant to multiple drugs. It is not negligible that the ongoing COVID-19 pandemic may further contribute to antimicrobial resistance. In this paper, multidrug resistance and antimicrobial resistance are underlined, focusing on the therapeutic options to overcome these obstacles in drug treatments. Lastly, some recent studies on nanodrug delivery systems have been reviewed since they may represent a significant approach for overcoming resistance.

## 1. Introduction

The development of simultaneous resistance to multiple drugs, with varying chemical structures and targets, is a major obstacle to effective cancer therapy [1,2]. Multidrug resistance (MDR) is a kind of acquired resistance of microorganisms and cancer cells to chemotherapic drugs that are characterized by different chemical structures and different mechanisms of action. MDR is the consequence of the overexpression of a variety of proteins that extrude the chemotherapic from the cell, lowering its concentration below the effective one. MDR in cancer treatment is responsible for tens of thousands of deaths per year and can be conferred by a number of transporters that pump drugs out of cells as, for instance, the adenosine triphosphate binding cassette (ABC) pumps. They can translocate a wide variety of substrates, including amino acids, peptides, ions, sugars, toxins, lipids, and drugs, and are implicated in several serious human diseases, including cystic fibrosis (CF) and several disorders of the immune system [3]. The ABC transporter family is a protein superfamily with 49 different members categorized by gene sequence and structural similarities. These ABC transporter families, expressed in various tissues such as the liver, intestine, kidney, and brain, are divided into seven subfamilies (ABCA to ABCG) based on their gene structure, amino acid sequence, domain organization, and phylogenetic analysis. Of these, at least 11 ABC transporters including P-glycoprotein (P-GP/ABCB1), multidrug resistance-associated proteins (MRPs/ABCCs), and breast cancer resistance protein (BCRP/ABCG2) are involved in multidrug resistance (MDR) development [4]. The human 170 kDa P-glycoprotein (P-gp, also referred to as multidrug resistance protein 1, MDR1, or ABCB1 or MDR1), the 190 kDa multidrug resistance-associated protein 1 (MRP1 or ABCC1), and the 70 kDa breast cancer resistance protein (BCRP or ABCG2) can transport diverse classes of amphipathic drug molecules. These three drug transporter pumps work through well-characterized mechanisms of MDR, knowledge of which has been exploited to find a winning approach to contrast the multidrug resistance, which is a significant hurdle in current cancer treatments and antimicrobial therapies. Thus, the inhibition of drug efflux pumps, such as P-gp, MRP1, and BCRP, has been pursued by several researchers. For instance, P-gp inhibition-based strategies for modulating pharmacokinetics of anticancer drugs have been recently reviewed [5]. The use of natural products has been proposed also as an alternative choice for P-gp inhibition [6]. Recently, the ABC superfamily has been classified into distinct types, I-VII, based on their transmembrane domain (TMD) fold [7]. Besides, bacteria can show resistance to one or more classes of antimicrobials and, on this base, they can be classified into: multidrug-resistant bacteria (i.e., resistant to three or more classes of antimicrobials), extensively drug resistant (i.e., resistant to all but one or two classes) or pandrug-resistant (i.e., resistant to all available classes). Antimicrobial resistance (AMR) to antibiotics is a growing global problem [8,9], which led to failure of even the most recent types of effective antibiotics [10] ensuring the need of a new molecule arsenal could no longer be postponed [11]. The ongoing Coronavirus Disease 2019 (COVID-19) pandemic and the lack of an effective therapeutic protocol could further contribute to AMR [12]. Indeed, with the massive, and sometimes inappropriate, use of antibiotics to treat COVID-19 and flu symptoms, AMR threat remains significant [13]. However, the relationship between COVID-19 and AMR is not clear. Guisado-Gil et al. (2020) [14] provided quantitative data about the pandemic effect on antimicrobial consumption, studying the impact of the COVID-19 pandemic in a tertiary care Spanish hospital with an active ongoing antimicrobial stewardship program (ASP). For a 20 week period, they monitored antimicrobial consumption, incidence density, and crude death rate per 1000 occupied bed days of candidemia and multidrug-resistant (MDR) bacterial bloodstream infections (BSI), and found that no change in the global trend of incidence of hospital-acquired candidemia and MDR bacterial BSI was observed (+0.5% weekly; *p* = 0.816). Tiri et al. (2020) [15] instead verified the bimonthly incidence of Carbapenem-Resistant Klebsiella pneumoniae (CR-Kp) (CRE) colonization patients and the incidence of CRE acquisition in an intensive care unit (ICU) during the period of January 2019 to June 2020. In Italy the infections due to antibiotic-resistant bacteria have largely attributed to CRE. The incidence of CRE acquisition went from 6.7% in 2019 to 50% in March–April 2020, despite the great attention and the training of all staff on infection control measures in the COVID-19 era. Moreover, drug repositioning, which consists of identifying and developing new uses for existing drugs, may represent a valid strategy for overcoming MDR and AMR [16,17]. The use of multi-target drugs with known toxicity profiles also proved to be a promising alternative for the treatment of bacterial infections and cancer [18,19]. Recently, nanomedicine, which represents a promising approach to improving drug efficacy and minimizing adverse effects, also turned out to be very useful in overcoming cancer drug resistance [20,21].

## 2. Efflux Pump System Mechanisms

Among the several mechanisms that induce antibiotic resistance, drug extrusion by the drug efflux pumps represents an important cause of MDR. Reduction in intracellular drug accumulation by increasing efflux of drugs and overexpressing the drug transporters has been associated with clinically relevant drug resistance. This represents the principal mechanism for the development of resistance to tetracyclines, erythromycin, and fluoroquinolones. It is also the main mechanism of resistance in cancer chemotherapy. Generally, the efflux pump systems are different in prokaryotic and eukaryotic organisms, even if some of their classes are able to mediate resistance in both the organisms [22]. The prokaryotic efflux pumps are divided into six classes, while the eukaryotic ones are divided into five groups. However, the ATP- binding cassette (ABC) efflux pumps are the major efflux pumps involved in multidrug resistance. The 48 human ABC transporter genes are classified into seven subfamilies (namely from ABCA to ABCG) [23]. Among them, P-glycoprotein is a well-known protein associated with multi-drug resistance. It belongs to the human ABCB (MDR/TAP) family and is also known as ABCB1 or MDR1 P-gp [24]. Structurally, ABC transporters are characterized by the presence of four domains consisting of two cytoplasmic nucleotide-binding domains (NBDs) that bind and hydrolyze ATP and two TMDs that identify and transport substrates [25]. In physiological conditions, ABC transporters control cellular levels of hormones, lipids, ions, xenobiotics, and other molecules by carrying them across cell membranes and are expressed in organs involved in elimination of these molecules as the kidney, liver, and epithelial tissues, and so on [26]. Among the 48 ABC transporters, some bind a wide range of substrates. Some of them also have the potential to transport anticancer or antimicrobial drugs, conferring drug resistance [27] Generally, ABC transporters pump the substrates against a chemical gradient, a process that necessitates ATP hydrolysis as a driving force. Under physiological conditions, ABC transporters operate in a single direction, even if, under certain conditions, the drug efflux pumps could be reversible. Several mechanisms of ABC transporter have been proposed, such as alternating site, the switch, and the constant contact models. However, it seems that all these models share common phases: first, the NBD dimerization ATP dependent and then the substrates binding to the TMD and its switching between outward- and inward-facing conformations [28]. A complete understanding of the mechanisms of drug transporters and the development of efflux pump inhibitors are crucial for promising anti-drug resistance solutions.

## 3. MDR in Cancer

Chemoresistance is thought to be the cause of mortality in more than 90% of patients with advanced cancer [29]. The first-line therapy is often followed by the proliferation of a small number of survived cancer cells that leads to the development of a secondary tumor that is insusceptible to the initial set of drugs. This may cause tumor progression after stabilization or significant regression because chemotherapy, which was successful at the first stage, becomes ineffective. Chemoresistance is also associated with metastases [30] and MDR occurs frequently after long-term chemotherapy, resulting in refractory cancer and tumor recurrence. Another important consideration is that cancer cells with acquired MDR often become cross-resistant to structurally unrelated chemotherapeutic drugs. Numerous cellular and non-cellular pathways have been proposed as theoretical mechanisms behind MDR, such as decreased uptake of water-soluble drugs, increased enzyme level of xenobiotic metabolism (e.g., glutathione-S-transferase), various changes in cells that affect the ability of cytotoxic drugs to kill them, and removal of hydrophobic drugs from cells due to increased energy-dependent efflux (Figure 1) [31]. ABC transporters are a superfamily of membrane proteins mediating MDR mechanisms in multiple types of cancers [32], as they are able to translocate hydrophobic drugs and lipids from the inner to the outer leaflet of the cell membrane. Among the ABC transporters involved in MDR, P-gp (MDR1, ABCB1) and MRP can be overexpressed in malignant cells and their activity results in lack of intracellular levels of the drug necessary for an effective therapy [33]. P-gp is the best-characterized efflux pump that mediates MDR in cancer [34], which targeting represents an interesting approach for combating multidrug resistance [23]. Subcellular expression of P-glycoprotein (P-gp) may play an essential role in multidrug resistance (MDR) in many cancers such as breast cancer cells [35], human colorectal cancer cells [36], ovarian cancer cells [37], human lung cancer [38,39], and others. Human ABCB1 transporter was the first recognized ABC transporter: its overexpression in cancer cells reduces the concentration of drugs in the cell and allows it to develop resistance to a number of chemotherapic drugs (Table 1), such as taxanes (paclitaxel), vinca alkaloids (vinblastine), and anthracyclines (daunorubicin) [40]. ABCB1 is a 170 kDa membrane transporter that is ubiquitously expressed in the kidney, intestine, brain, and placenta. Clinical applications, exploiting the combination of an ABCB1 modulator and an anticancer drug, have been investigated as a possible strategy to overcome ABCB1-mediated drug efflux for a long time [41]. Interestingly, some studies reveal the potential role of melatonin in chemotherapeutic synergy for countering MDR. Melatonin has been reported to have an influence on the P-gp expression, increasing the chemotherapy sensitivity of colon cancer cells. In particular, the authors examined the effect of melatonin on LoVo colon cancer cells’ resistance to doxorubicin. They found that some concentrations of MLT and DOX increased the percentage of cells expressing P-gp [42]. As explanatory example, epirubicin is a first-line chemotherapeutic drug for the clinical treatment of diffuse large B cell lymphoma (DLBCL), but the overexpression of MDR transporter proteins, especially P-gp, determines its ineffectiveness [43]. The contemporaneously use of melatonin has been demonstrated to downregulate the expression of P-gp, ultimately sensitizing DLBCL cells to epirubicin that suppresses their growth [43,44]. Few developed inhibitors of P-gp have been approved for use in cancer therapy because of a lack of significant clinical efficacy, or concerns about their clinical safety. A new emerging class is represented by the antimicrobial peptides, including naturally occurring peptides as well as their synthetic derivatives that have the ability to respond to infections, receiving attention to make resistant strains sensitive to existing drugs [45]. Recently, the antimicrobial peptide XH-14C was suggested in combination with conventional anticancer agents as a novel strategy for cancer treatment. It was found that XH-14C reverse ABCB1-mediated MDR, enhancing the intracellular accumulation of the paclitaxel—an ABCB1 substrate-drug—by directly inhibiting the efflux function of ABCB1 without affecting the transporter’s expression and cellular localization [46]. Another important member of the ABC transporters is the ABC transporter subfamily G member 2 (ABCG2), which is also named BCRP. It is present in the apical membranes of many epithelial cells and tissues, including lung, gut, intestine, liver, breast, placenta, hematopoietic stem cells, and especially in the blood–brain barrier. It is overexpressed in many solid tumors as well as acute myeloid leukemia (AML) and acute lymphocytic leukemia (ALL). Dysregulated ABCG2 overexpression is linked with poor prognosis in several cancer types with particularly low survival in AML patients. High ABCG2 expression in cancer cells results in resistance to a wide spectrum of chemotherapeutic agents, including mitoxantrone, topotecan, SN-38, and doxorubicin [47]. Numerous studies have shown that the tumors expressing high levels of ABCG2 are refractory to anticancer drugs [48]. Recent studies suggest a synergistic effect of Venetoclax (Venclexta, Venclyxto, ABT-199, GDC-0199, RG7601) with certain chemotherapeutic agents for the treatment of MDR cancers. Venetoclax is a potent and selective Bcl-2 inhibitor, approved by the FDA in 2016 for the treatment of patients with chronic lymphocytic leukemia (CLL) with 17p deletion. Bcl-2 represents an antiapoptotic protein playing an important role in tumorigenesis and chemoresistance, which is often overexpressed in hematopoietic malignancies. Venetoclax was demonstrated to enhance the efficacy of ABCG2-substrate anticancer agents by directly inhibiting ABCG2-ATPase activity, blocking the efflux function of wild-type ABCG2, and therefore increasing the intracellular accumulation of the chemotherapeutic drugs. A docking simulation suggested that Venetoclax could bind the drug-binding pocket and ATP-binding site of ABCG2, impeding its efflux activity [49]. The development of novel reversal agents that could inhibit the efflux functions of ABC transporters is an urgent need that could bring some advantages in chemotherapy. Several new potential drugs for the treatment of cancer are under study [50,51,52] and the complexes with transition metals represent an important alternative way to combat MDR [53,54,55,56]. The modulation of reactive oxidative species (ROS) may represent another strategy to kill MDR cancer cells that are mechanistically diverse [57,58]. In cancer cells, ROS are involved in the regulation and induction of apoptosis, thereby they play a central role in modulating cancer cells proliferation, survival, and drug resistance. The ROS levels and the activity of scavenging/antioxidant enzymes in drug resistant cancer cells are higher than those found in non-MDR cancer and normal cells [59]. Therefore, MDR cancer cells may be more susceptible to alterations in ROS levels, thus compounds able to modulate ROS levels can exert a sensitizing effect and enhance the response to certain chemotherapeutic drugs. In this context, natural products, such as *Salvia fruticosa* Mill. and pomegranate *Akko* peel, thanks to their demonstrated antioxidant and antiproliferative activities [60,61], could represent a valid approach useful for overcoming cancer resistance. Indeed, it is noteworthy that the polyphenol resveratrol has been proposed as a MDR reversion molecule in breast cancer and its derivatives exhibited good anticancer properties, inducing apoptosis [62,63]. In addition, the autophagy pathways seem to be involved in the development of MDR [64]. This is a self-degradative system that arises during the treatment of sensitive and MDR cancer and has a double face in MDR tumors, as it participates in the development of MDR and protects cancer cells from chemotherapeutics. However, some results indicate that autophagy induced by anticancer drugs may kill MDR cancer cells for which apoptosis pathways are inactive or, alternatively, determine the activation of apoptosis signaling pathways, thus facilitating MDR reversal [65]. Another way to combat MDR exploits the inhibition of P-gp and p53-Mdm2 protein–protein interaction (PPI), since the p53-Mdm2 pathway is compromised in more than 50% of all human cancers. p53-Mdm2 PPI inhibitors were suggested as a promising platform for the development of multitarget drugs that can overcome tumor resistance by inhibiting the P-gp activity [66].

## 4. MDR and Antimicrobials

The rapid development of MDR bacteria due to the improper and overuse of antibiotics, together with their ineffective performance, against the difficult-to-treat biofilm-related infections (BRIs) have urgently called for alternative antimicrobial agents and strategies to combat bacterial infections [67]. MDR in bacteria may be generated by several mechanisms. First, bacteria may accumulate multiple genes—each coding for resistance to a single drug—within a single cell, and this accumulation typically occurs on resistance R-plasmids. Moreover, multidrug resistance may also occur due to the increased expression of genes that code for multidrug efflux pumps, extruding a wide range of drugs. Finally, MDR can be developed by enzymatic inactivation of the drugs through their degradation or by transfer of a chemical group to them. Some drugs can be inactivated by hydrolyzation (penicillin, tetracycline, etc.). Drug inactivation by transfer of a chemical group commonly occurs through the transfer of acetyl, phosphoryl, and adenyl groups [68]. Antimicrobial resistance of non-fermenting Gram-negative bacteria is increasingly recognized as an urgent healthcare threat and has been reported from different areas all over the world [69]. MDR may also refer to clinically important multi-resistant Gram-positive bacteria, such as *Enterococcus faecium* and *E. faecalis* [70]. In-depth studies on bacteria could be essential to fully understand the physiological functions of these microbes and consequently overcome problems related to MDR [71,72]. The search for new antimicrobial agents that may overcome AMR is a very important goal to pursue [73,74]. Recent literature describes many compounds with antimicrobial activity [75,76,77]. Antimicrobial peptides (AMPs), owing to their compelling antimicrobial activity against MDR bacteria and BRIs without causing bacteria resistance, are promising alternative antimicrobial agents to combat MDR [78]. For instance, the antibiotic-resistant *Pseudomonas aeruginosa* infections are the primary cause of mortality in people with CF [79]. The antimicrobial peptide DP7, designed in silico, possesses a broad-spectrum antimicrobial activity by inhibiting the growth of clinical *P. aeruginosa* strains and reducing biofilm formation. In acute lung infection, it exhibited a 70% protection rate and reduced bacterial colonization by 50% in chronic infection. It mainly suppressed gene expression involving lipopolysaccharide (LPS) and outer membrane proteins and disrupted cell-wall structure [80]. Recently, a great deal of effort has been directed towards the problem of fighting against biofilm formation by bacteria. A biofilm is closed layer of bacteria that are adherent to each other, forming a polymer matrix consisting of polysaccharide, protein, and DNA. Bacterial biofilms increase tolerance to antibiotics and disinfectant and their survival is ensured by chromosomal beta-lactamase, upregulated efflux pumps, and mutations in antibiotic target molecules in bacteria [81]. Multidrug-resistant *Enterobacterales* (MDR-E) strains to carbapenems and other extended-spectrum-β-lactam may cause colonization or infection in solid-organ transplantation (SOT) recipients with mortality rates ranging from 5 to 20% [82]. A phage therapy has been recently proposed. Phages, viruses that infect bacteria, are environmentally ubiquitous, host-specific, and effective at infecting MDR bacterial strains [83,84]. The phage proposed therapy employs OMKO1, a lytic bacteriophage (family *Myoviridae*) of *P. aeruginosa* that utilizes the outer membrane porin M (OprM) of the multidrug efflux systems MexAB and MexXY as a receptor-binding site. This may represent a new approach to phage therapy where bacteriophages exert selection for MDR bacteria to become increasingly sensitive to the traditional antibiotics [85]. This therapy could not only improve the clinical efficacy against MDR bacteria, but also potentially slow or reverse the incidence of antibiotic resistant bacterial pathogens [86]. Recently, mutations of the *mexEF* and *mexR* efflux pump systems in *P. aeruginosa*, due to a treatment with the antibiotic aztreonam, have been investigated in acute murine lung infection. However, literature data reported that, even though *mexR* mutations are common in CF [87], the frequency of mutants having both mutations is still unclear, and the murine model employed is not very similar to the chronic infections affecting people with CF [88]. Pandrug-resistant *Klebsiella pneumoniae* (PDR-Kp) may cause bacteremia with high mortality, especially among patients with septic shock [89]. Multidrug-resistant tuberculosis (MDR-TB) is a threat for the global TB epidemic control in adults and children and, in 2015, 10 million new cases were reported worldwide [90]. MDR-TB is defined as simultaneous resistance to rifampicin and isoniazid, the cornerstones of the treatment of drug-susceptible tuberculosis, necessitating the use of expensive and toxic second-line treatment regimens. For instance, amikacin has been used for more than 40 years, although controversy over the right dose remains [91]. Delamanid and Bedaquiline are used, also in association, in MDR-TB [92]. A WHO (World Health Organization) strategy for worldwide eradication of tuberculosis is the directly observed therapy short-course (DOTS), followed by therapy [93], and an update by the Global Tuberculosis Network has been recently reported [94]. The overexpression of drug efflux pumps belonging to the ABC superfamily is also a frequent cause of resistance to antifungals [95]. MDR in yeasts seems to be related to pleiotropic drug resistance (PDR) subfamily and major facilitator superfamily (MFS) transporters [96] even though their mechanisms remain unclear [97,98]. Azole resistance is a major concern for treatment of infections with *Aspergillus fumigatus*, a saprophytic mold that can cause a range of clinical syndromes ranging from allergic conditions to acute and chronic invasive pulmonary aspergillosis, especially in immunosuppressed patients [99]. The recent emergence of *Candida auris* has caused significant concerns, given its worldwide distribution and high reported incidence of antifungal resistance, moreover it has been estimated that 93% of clinical isolates exhibit increased resistance to fluconazole [100]. Recent studies about MDR in microbes regards a new type II topology of ABC transporters, the *Candida* drug resistance 1 and 2 proteins, Cdr1p and Cdr2p in *C. albicans*, and the Pleiotropic drug resistance 5 protein Pdr5p in *Saccharomyces cerevisiae* [101]. Finally, it is becoming recognized that resistance to antibiotics can occur either by mutations or by acquisition of resistance conferring genes via horizontal gene transfer (HGT). Multiple mechanisms of HGT are discovered: conjugation by plasmids, transduction by bacteriophages, and natural transformation by extracellular DNA allow genetic material to jump between strains and species. Thus, HGT contributes significantly to the rapid spread of resistance, even if the transmission dynamics of genes, that confer antibiotic resistance are not completely understood [102,103].

## 5. MDR and Antivirals

MDR in viruses is another serious problem of the last century and it is often related to MDR in bacteria, as happens in patients receiving treatments for MDR-TB and hepatitis C virus (HCV) or human immunodeficiency virus (HIV) [104,105]. Bacteria and other microorganisms have evolved several different resistance mechanisms, while resistance to antivirals occurs only as a result of mutations in the genes that encode antiviral target sites or antiviral drug activators [106]. Indeed, antiviral resistance usually involves amino acid substitutions in the target protein that prevent drug binding or prevent an enzyme from accepting the drug as a substrate [107]. The emergence of a multidrug-resistant pandemic influenza A (H1N1) virus was reported in 2010 in a patient treated with neuraminidase inhibitors, with a novel resistance pattern that conferred resistance to oseltamivir, zanamivir, and peramivir [108]. Moreover, studies in patients with multidrug-resistant HIV have been also carried out [109]. Recently, the acquisition of MDR HIV-1 infection in a patient taking pre-exposure prophylaxis with a combination tenofovir disoproxil fumarate and emtricitabine has been reported [110]. Treatment with fostemsavir, the prodrug of the HIV-1 attachment inhibitor temsavir, has been suggested as a valuable therapeutic option in heavily treatment-experienced patients harboring MDR virus, with limited therapeutic options [111]. However, new strategies for obtaining effective antivirals are needed [112]. MDR to antibiotics is a growing worldwide problem to which the ongoing COVID-19 pandemic may further contribute. With resources deployed away from antimicrobial stewardship, evidence of substantial pre-emptive antibiotic use in COVID-19 patients and indirectly, with deteriorating economic conditions fueling poverty and potentially impacting on levels of resistance, AMR threat remains significant [113]. Bork et al. (2021) [114] evaluated the effects of antibiotic use in the multidrug-resistant Gram-negative (MDRGN) acquisition among COVID-19 patients in an academic hospital. They found that MDRGN acquisition increased 3% for every increase in positive COVID-19 tests per week. The high antibiotic utilization may have contributed to the increase in MDRGNs among COVID-19 patients, together with other factors (altered infection control practices, critical illness and prolonged hospital stay of COVID-19 patients) [114]. Ukuhor (2021) reported that various types of bacterial and fungal infections occur in patients with COVID-19 with some resistant to antimicrobials that are associated with significantly worse outcomes and deaths [115]. In another study of Lobie et al. (2021) [115] is reported that some disinfectants, such as sanitizers, contain genotoxic chemicals that damage microbial DNA, activating error-prone DNA repair enzymes, which can lead to mutations that induce antimicrobial resistance. On the other hand, it is fair to report a retrospective study that demonstrated a reduction in MDR bacterial infections during the COVID-19 pandemic. The authors assess that a high level of preventive measures could help tackle an important health problem such as that of the spread of MDR bacteria [116]. Moreover, increased hand hygiene, decreased international travel, and decreased elective hospital procedures may reduce AMR pathogen selection and spread in the short term [117]. However, the lack of studies supporting one hypothesis or the other one makes any deduction far from a sound scientific interpretation.

## 6. Nanocarriers

Nanotechnology-based chemotherapies are gaining increasing interest in cancer drug treatment, with particular attention to MDR cases. Indeed, nanocarriers such as liposomes, polymeric micelles, metallic nanoparticles, or dendrimers all contribute to co-delivering different chemotherapeutics, and may represent an interesting strategy to increase the drug selectivity, diminish the onset of undesired side effects and, last but not least, overcome the resistance in cancer [118,119]. Increasing efforts in the development of new materials able to confer selectivity towards cancerous tissues to antineoplastic compounds is essential. The selective accumulation of chemotherapeutics in the tumor region can be obtained with a combination of active and passive targeting that can further help to reduce the toxic effects to healthy tissues/organs and to overcome the resistance phenomena [120]. Gold nanoparticles (AuNPs) have attracted scientific interests among various nanocarriers developed for use in nanomedicines and have been actually used to overcome cancer MDR against chemotherapeutic agents in cancer cells [121,122]. As example, Doxorubicin was coupled with PEGylated AuNPs of 5 nm through enzyme-cleavable disulfide linkage or conjugated to AuNPs through a polyethylene glycol (PEG) spacer using acid labile hydrazine linkage [122]. Generally, the modification of biological molecules by covalent conjugation with PEG, PEGylation, improves drug solubility and decreases immunogenicity. PEGylation also increases drug stability and the retention time of the conjugates in blood, resulting in an improvement in the pharmacokinetic behavior of the drug [123]. The two obtained nanocargo systems were investigated for reversing MDR in HepG2-R MDR cell line and breast cancer MCF-7/ADR cell line, respectively. The drug anticancer activity was improved significantly in the MDR cancer cells upon conjugating to the designed AuNPs in both cases [124]. The size of AuNPs is crucial and the critical size for overcoming MDR was identified to be between 4.1 and 5.4 nm [125]. The use of nanomaterials for antibacterial activity, the type of nanomaterials, the strategies to tackle their toxicity, as well as their limits, have been recently reviewed. Generally, the nanoparticles (NPs) can overcome the common resistance mechanisms, such as enzymatic inactivation or modification of the active site, diminished cell permeability or increased activity/expression of efflux pumps, allowing the antibacterial activity. In addition, if NPs are conjugated with antibiotics, they show synergistic effects against inhibiting the biofilm formation in multidrug-resistant organisms [126,127,128]. The employment of NPs is also widely studied to overcome multidrug resistance for the treatment of methicillin-resistant *Staphylococcus aureus* (MRSA), a Gram-positive bacterium responsible for many complicated infections, including those affecting skin and soft tissue, bone and joints together with pneumonia, osteomyelitis, and infective endocarditis [129]. Nanodrug delivery systems have great potential in the translation of MRSA treatments towards better therapeutic outcomes and represent a significant approach for overcoming bacterial MDR. Finally, the nano-biomaterials have been suggested for the treatment of MDR-TB [130]. AMPs-based nano-formulations have significantly improved the therapeutic effects of AMPs in various bacterial infections models, including bloodstream (specifically sepsis), pulmonary, chronic wound, and gastrointestinal infections, by ameliorating their hydrolytic stability, in vivo half-life, and solubility, as well as reducing the cytotoxicity and hemolysis [131]. In summation, the most promising perspective on using nanotechnology resides in the possibility to wrap several types of drugs in nanocarriers and/or connect even different chemical substituents and specific substances on their surface to achieve optimal targeting, increase the drug stability and selectivity, and decrease the risk of undesired effects. Unfortunately, some disadvantages should be considered, because the in vivo NPs pharmacokinetics and excretion are not yet fully understood and require further study.

## 7. Summary

The MDR phenomenon represents a serious medical problem and constitutes a major challenge to the treatment of infectious diseases and cancer and the development of novel therapeutics. MDR in cancer cells is mainly due to drug efflux transporters, particularly those belonging to the ABC superfamily. Different treatments of choice are available for the MDR, among which the most common are represented by the inhibition of P-gp, MBR, and ABCG2, and the modulation of ROS. However, given the adverse effects of these drugs, investigating and establishing effective and non-toxic drugs to reverse MDR in cancers has become a pressing need. Moreover, the increasing prevalence and severity of MDR bacterial infections has necessitated novel antibacterial strategies. Amongst them, nanotechnology has proved to be a promising tool for the whole scientific community by creating new therapies for advanced treatment of various diseases. The design of nanosystems for co-delivering anticancer agents may represent a new challenge to combat resistance, although further in vivo studies regarding their pharmacokinetics and metabolism are necessary.

## Figures and Tables

**Figure 1 molecules-27-00616-f001:**
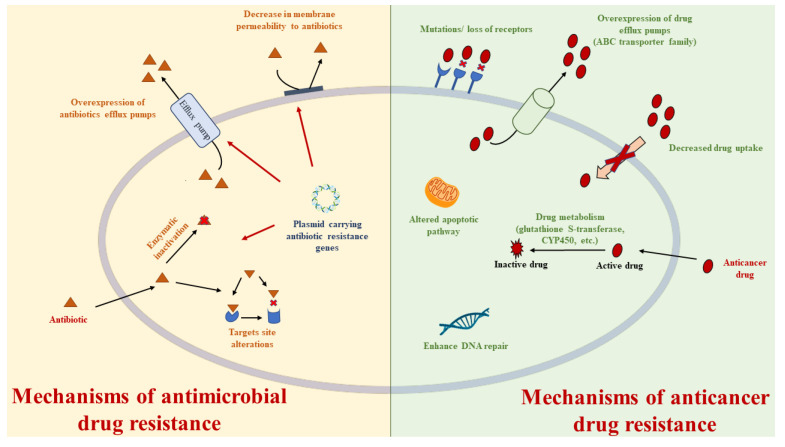
Mechanisms of antimicrobial and anticancer drug resistance.

**Table 1 molecules-27-00616-t001:** Structure of drugs mentioned in the text.

Formula	Name
** *Anticancer Drugs* **	
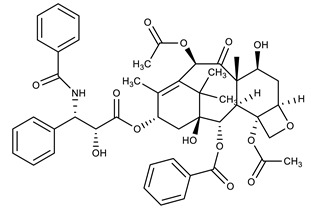	Paclitaxel
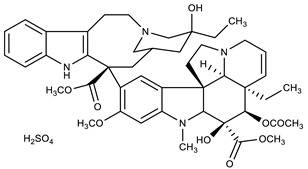	Vinblastine
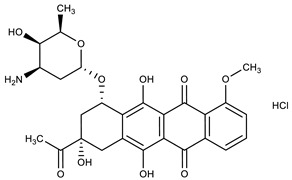	Daunorubicin
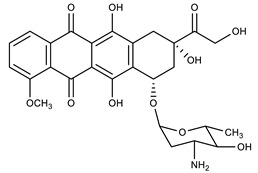	Epirubicin
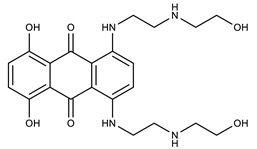	Mitoxantrone
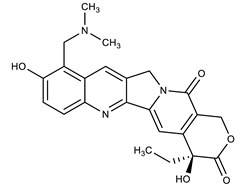	Topotecan
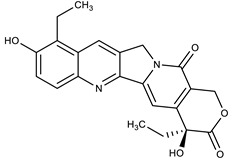	SN-38
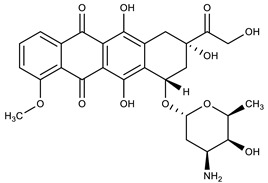	Doxorubicin
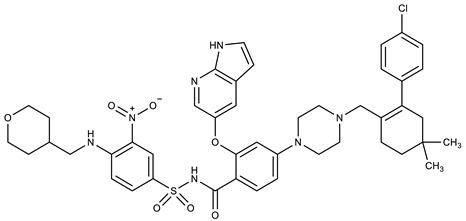	Venetoclax
** *Antimicrobial drugs* **	
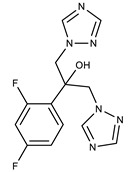	Fluconazole
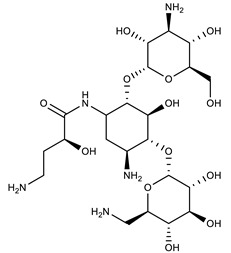	Amikacin
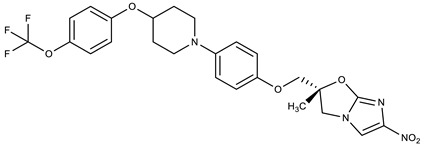	Delamanid
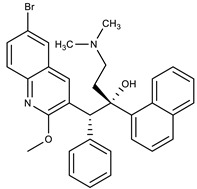	Bedaquiline
** *Antiviral drugs* **	
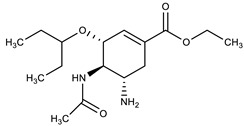	Oseltamivir
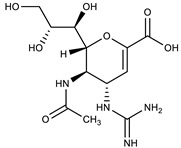	Zanamivir
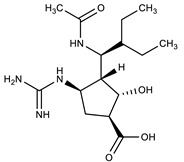	Peramivir
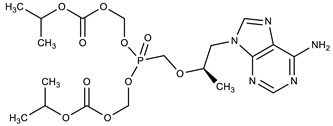	Tenofovir
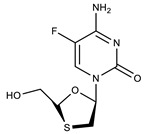	Emtricitabine
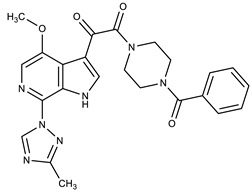	Temsavir

## Data Availability

Not applicable.

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
