# Peer review of "Multidrug Resistance (MDR): A Widespread Phenomenon in Pharmacological Therapies"

_molecules, 2022, doi:10.3390/molecules27030616_

Round 1

Reviewer 1 Report

The authors review the role of multi-drug resistance in cancer and microbes, along with the use of antimicrobial proteins and nanoparticles as means to overcome MDR. It is a very relevant topic with broad implications reaching larger audiences. While the review contains specific examples, structure is largely missing and they appear to have been put together without much prior organization. The authors touched upon several topics without discussing each topic in detail. With just one paragraph for each section, it is hard to even notice the transition between individual subtopics. I suggest dividing each section into different paragraphs, if not different subsections to provide a structure for the review. Please find below some of the specifics.

Pg. 2 second sentence: It appears that there are 49 families of ABC transporters, in fact, there are 49 ABC genes classified into 8 subfamilies. Can authors clarify this?

line 73: although authors acknowledge a lack of coherent correlation between COVID-19 and AMR, it is not clear what the authors are trying to convey here. For example, what is the rationale behind the reduction in AMR bacterial infection during the current pandemic, also include a relevant citation.

It would be good to include a section on the mechanism by which drug transporters work, including the classification of ABC transporters before going into drug resistance in cancer cells.   

line 99: “MRP can” is rather elusive – be specific on what cell lines overexpress MRPs.

Line 111: “contrasting”? is it countering?

Line 112: “influence” – again very elusive. Be specific on how it modulates MRP.

Line 115: provide appropriate reference to corroborate the sentence.

Line 122: again “could” is inconclusive, suggest if XH-14C inhibits ABCB1-mediated MDR. The use of antimicrobial peptides as an anti-cancer agent is significant and a new line of therapy. Can authors discuss in detail the significance and discuss the role of antimicrobial peptides in modulating drug pumps.

Line 195: discuss the role of biofilm in drug resistance in bacteria.

Line 204: “proposed therapy” instead of “therapy proposed”?

Line 211: can the authors discuss in detail, the use of antibiotics as the cause of mutations on the efflux pump?

Line 243: can authors provide a potential mechanism for the development of drug resistance in viruses?

Line 256: while it is understandable how preemptive use of antibiotics resulting in their overuse could potentially lead to MDR bacterial strains, can authors provide specifics if there are any? This appears hypothetical in the current format.

Line 266: Nanocarriers – can authors provide the mechanism by which nanoparticles provide cell-type selectivity? Also, what is the significance or PEGylation of AuNps?

Figure1: Not sure if enzymatic inactivation for anti-microbials was explained?

Author Response

The reply to reviewer 1 are in the enclosed file.

Reviewer 2 Report

This is a well written review manuscript considering multi drug resistance from a broad perspective, including cancer, bacterial, mold and viral infections. However, the authors have not paid enough attention to horizontal gene transfer as a main mechanism for bacterial resistance spread (not limited to plasmids). Please introduce additional information about this item.

Minor points:

  • The Figure including all the structures of the compounds mentioned in the manuscript must be divided in 3 parts (3 figures or sub-figures): (i) anticancer compounds; (ii) antibacterial compounds; (iii) antiviral compounds.
  • L237: Candida must be in italics.

Author Response

The replies to reviewer 2 are in the enclosed file.

Round 2

Reviewer 1 Report

The changes made to the paper are satisfactory.